# Shape-Tailored Deep Neural Networks With PDEs

**Naeemullah Khan**
Department of Engineering Science
University of Oxford
naeemullah.khan@eng.ox.ac.uk

**Angira Sharma**
Department of Computer Science
University of Oxford
angira.sharma@cs.ox.ac.uk

**Philip H. S. Torr**
Department of Engineering Science
University of Oxford
philip.torr@eng.ox.ac.uk

**Ganesh Sundaramoorhti**
Raytheon Technologies
ganesh.sundaramoorthi@rtx.com

## Abstract

We present Shape-Tailored Deep Neural Networks (ST-DNN). ST-DNN are deep networks formulated through the use of partial differential equations (PDE) to be defined on arbitrarily shaped regions. This is natural for problems in computer vision such as segmentation, where descriptors should describe regions (e.g., of objects) that have diverse shape. We formulate ST-DNNs through the Poisson PDE, which can be used to generalize convolution to arbitrary regions. We stack multiple PDE layers to generalize a deep CNN to arbitrarily shaped regions. We show that ST-DNN are provably covariant to translations and rotations and robust to domain deformations, which are important properties for computer vision tasks. We show proof-of-concept empirical validation.

## 1   Introduction

Poisson Partial differential equations (PDEs) have desirable robustness and covariance properties and can be tailored to region of interest, which are not present in modern convolutional neural networks (CNNs). In this work we try to bridge this gap by introducing PDEs in CNNs architecture. To show the advantage of these properties, we show their utility in segmentation.

CNNs have been used extensively for segmentation problems in computer vision (10; 11; 6; 23). CNNs provide a framework for learning descriptors that are able to discriminate different textured or semantic regions within images. Much progress has been made in segmentation with CNNs but results are still far from human performance. Also, significant engineering must be performed to adapt CNNs to segmentation problems. A basic component in the architecture for segmentation problems involves labeling or grouping dense descriptors returned by a backbone CNN. A difficulty in grouping these descriptors arises, especially near the boundaries of segmentation regions, as CNN descriptors aggregate data from fixed shape (square neighborhoods) at each pixel and may thus aggregate data from different segmentation regions. This makes grouping these descriptors into a unique region difficult, which often results in errors in the grouping.

Our contributions are specifically (see Figure 1): 1) We construct and show how to train ST-DNN, deep networks that perform shape-tailored spatial filtering via the Poisson PDE at each depth so as to generalize a CNN to arbitrarily shaped regions. 2) We show analytically and empirically that ST-DNNs are covariant to translations and rotations as they inherit this property from the Poisson PDE. In segmentation, covariance (a.k.a., equivariance) to translation and rotation is a desired property: if a segment in an image is found, then the corresponding segment should be found in the translated / rotated image (or object). This property is not generally present with existing CNN-based

segmentation methods even when trained with augmented translated and rotated images (3), and requires special consideration.3) We show analytically and empirically that ST-DNNs are robust to domain deformations. These result from viewpoint change or object articulation, and so they should not affect the descriptor. 4) To demonstrate ST-DNN and the properties above, we validate them on the task of texture segmentation, an important problem in low-level vision (17; 2).

Because of properties of the PDE, ST-DNN also have desirable generalization properties. This is because: a) The robustness and covariance properties are built into our descriptors and do not need to be learned from data, b) The PDE solutions, generalizations of Gabor-like filters (18; 26), have natural image structure inherent in their solutions and so this does not need to be learned from data, and c) Our networks have fewer parameters compared to existing networks in segmentation. This is because the PDE solutions form a basis and only linear combinations of a few basis elements are needed to learn discriminative descriptors for segmentation. In contrast, CNNs spend a lot of parameters to learn this structure.

## 1.1 Related Work

CNNs have been applied to compute descriptors for semantic segmentation, where pixels in an image are classified into certain semantic object classes (15; 12; 8; 20; 27; 16). Usually these classes are limited to a few object classes and do not tackle general textures, where the number of classes may be far greater, and thus such approaches are not directly applicable to texture segmentation. But semantic segmentation approaches may eventually benefit from our methodology as descriptors aggregating data only within objects or regions are also relevant to these problems. In (14) a learned shape-tailored descriptor is constructed with a Siamese network on hand-crafted shape-tailored descriptors. However, (14) only does shape-tailored filtering in pre-processing as layering these requires new methods to train. We further examine covariance and robustness, not examined in (14). Deformable convolutions (7) can also learn non rectangular receptive fields; however, they are not provably robust and covariant, key motivations of our approach.

Covariance to rotation in CNNs has been examined in recent works, e.g., (22; 25; 1). They, however, are not shape-tailored so do not aggregate data only within shaped regions. Lack of robustness to deformation (and translation) in CNNs is examined in (3) and theoretically in (4). (21) constructs deformation robust descriptors inspired by CNNs, but are hand-crafted.

## 2 Shape-tailored DNN through Poisson PDE and it's Properties

In this section, we design a deep neural network that outputs descriptors at each pixel within an arbitrary shaped region of interest and aggregates data only from within the region.

**Shape-tailored Smoothing via Poisson PDE**: To construct a shape-tailored deep network, we first smooth the input to a layer using the Poisson PDE so as to aggregate data only within the region of interest. Let $R \subset \Omega \subset \mathbb{R}^2$ be the region of interest, where $\Omega$ is the domain of the input image $\mathbf{I} : \Omega \to \mathbb{R}^k$ and $k$ is the number of input channels to the layer. Let $\mathbf{u} : R \to \mathbb{R}^M$ ($M$ is the number of output channels) be the result of the smoothing; the components $u$ of $\mathbf{u}$ solve the PDE within $R$:

$$\begin{cases} u(x) - \alpha \Delta u(x) = I(x) & x \in R \\ \nabla u(x) \cdot N = 0 & x \in \partial R \end{cases}, \tag{1}$$

where $I$ is a channel of $\mathbf{I}$, $\partial R$ is the boundary of $R$, $N$ is normal to $\partial R$, $\alpha$ is the scale of smoothing and $\Delta / \nabla$ are the Laplacian and the gradient respectively. It can be shown that the smoothing can be written in the form $u(x) = \int_R K(x, y) I(y) dy$ where $K(.,.)$ is the Green's function of the PDE, a smoothing kernel, which further shows that the PDE aggregates data only within $R$ (see Figure 1).

**Shape-tailored Deep Network**: We can now generalize the operation of convolution tailored to the region of interest by taking linear combinations of partial derivatives of the output of the PDE equation 1. This is motivated by the fact that in $R = \mathbb{R}^2$, linear combinations of derivatives of Gaussians can approximate any kernel arbitrarily well. Gaussian filters are the solution of the heat equation, and the PDE equation 1 relates to the heat equation, i.e., equation 1 is the steady state solution of a heat equation. Thus, linear combinations of derivatives of equation 1 generalize convolution to an arbitrary shape $R$; in experiments, a few first order directional derivatives are sufficient for our segmentation tasks (see Section 4 for details). A layer of the ST-DNN takes such

linear combinations and rectifies it as follows:

$$f_i[I](x) = r \circ L_i \circ T[I](x), \tag{2}$$

where $I : \mathbb{R} \to \mathbb{R}^k$ is the input to the layer, $T$ is an operator that outputs derivatives of the solution of the Poisson PDE equation 1, $L_i(y) = w_i y + b_i$ is a point-wise linear function (i.e., a $1 \times 1$ convolution applied to combine different channels), $r$ is the rectified linear function, and $i$ indexes the layer of the network. Notice that since $r$ and $L_i$ are pointwise operations, they preserve the property of $T$ that it aggregates data only within the region $R$. We now compose layers to construct a ST-DNN as follows:

$$F[I](x) = s \circ f_m \circ f_{m-1} \circ f_{m-2} \circ .... f_0 \circ I(x), \tag{3}$$

where $F[I](x)$ is the output of the ST-DNN, $f_0, ..., f_m$ are the $m+1$ layers of the network, $I$ is the input image, and $s$ represents the soft-max operation (to bound the output values).

ST-DNN does not have a pooling layer because the PDE already aggregates data from a neighborhood by smoothing; further, the lack of reduction in spatial dimension allows for more accurate shape estimation in our subsequent segmentation, and avoids the need for up-sampling layers. We will show that we can retain efficiency in training and inference.

## 2.1  Covariance and Robustness of ST-DNN

In addition to ST-DNN generalizing CNNs to arbitrary shaped regions, the ST-DNN is also covariant to in-plane translation and rotation, and robustness to domain deformations due to properties of the Poisson PDE. This means covariance also extends to our segmentation method, which is important as any object segmented in an image will also be segmented if the camera undergoes these transformations. Robustness to deformations is important as this means that small geometric variability (e.g., shape variations in textons, small viewpoint change, object deformation) will not affect the descriptors and the segmentation. We make these properties more precise, and give intuition for proofs, leaving details to Appendix (see 5.1).

**Definition 1** *An operator $S : \mathcal{I} \to \mathcal{I}$ (from the set of images $\mathcal{I}$ to itself) is* **covariant** *to a class $\mathcal{W}$ of transformations if $S[I \circ w] = [SI] \circ w$ for every $I \in \mathcal{I}$ and $w \in \mathcal{W}$.*

**Theorem 1** *The ST-DNN equation 3 is covariant to the set of translations and rotations, i.e., $x \to \mathcal{R}x + \mathcal{T}$ where $\mathcal{R}$ is a $2 \times 2$ rotation matrix and $\mathcal{T} \in \mathbb{R}^2$.*

**Theorem 2** *The ST-DNN equation 3 is insensitive to deformations, i.e.,*

$$|F[I \circ w] - F[I]| \le C\|w - id\|_{H^1}, \tag{4}$$

*where $w : \Omega \to \Omega$ is a domain deformation, $id$ is the identity map, $H^1$ is the Sobolev norm (measures both the amount and smoothness of the deformation), and $C$ is a constant independent of $w, I$.*

## 3  Training and Inference with ST-DNNs

Given the ST-DNN of Section 2, the loss function to train such descriptors from ground truth segmentation masks (motivated by consistency to the segmentation algorithm based on classical segmentation energies (5; 24)) is defined as:

$$L(\mathbf{W}) = \sum_{i=1}^{N} \frac{1}{|R_i|} \int_{R_i} ||\mathbf{F}_{\mathbf{W}}(x) - \mathbf{a_i}||_2^2 dx - \sum_i \sum_{j \ne i} ||\mathbf{a_i} - \mathbf{a_j}||_2^2 \tag{5}$$

where $i, j \in \{1, 2, ..., N\}$ are the indices for the regions in the ground truth segmentation, $\mathbf{F}_{\mathbf{W}}(x)$ is the output of the ST-DNN, $\mathbf{W}$ are the weights of the network (i.e., weights on derivatives of the Poisson PDE solution), $|R_i|$ is the area of region $i$, and $\mathbf{a_i}$ is the average descriptor within the $i^{\text{th}}$ region, i.e., $\mathbf{a_i} = \frac{1}{|R_i|} \int_{R_i} \mathbf{F}_{\mathbf{W}}(x) dx$. The loss function is comprised of two terms. The first component of the loss is minimized when the learned descriptor is constant within regions $R_i$ so that each region consists of parts of the image with uniform descriptor. The second term forces the learned descriptor of different regions to be different to discriminate different textures.

Computing gradients of the loss function for training requires consideration as it involves differentiating through PDEs. The most straightforward way to do this involves discretizing the PDE, so the solution is a linear matrix system as we do in this paper. This allows the use of existing deep learning packages to perform backprop by storing the matrix in memory. However, this can lead to large memory consumption as the matrix can be large and is only feasible for small images. The more accurate method, though more difficult to implement, is to avoid storing the matrix and instead compute the solution by an iterative numerical PDE method that does not require storage of matrices.

At inference the regions of segmentation are estimated iteratively together with updates of the ST-DNN for each of the regions as they evolve. We minimize the (non-convex) energy i.e. the first term of the training loss plus boundary regularisation term, with respect to the region. We use gradient descent to minimize the energy. The curve (boundary of regions) evolution is implemented with a method analogous to level set methods (19) by evolving smooth indicator functions of regions for convenient implementation. The method involves joint updates of the regions and the shape-tailored descriptors within the evolving regions. Our method typically takes a few iterations (approx. 20) to converge in our experiments. For more details of the inference refer to Appendix Algorithm 1.

## 4   Experiments

For the experiments we have used a four layer ST-DNN with 100, 40, 20, 5 hidden units respectively. The smoothing parameter $\alpha$ is set to 5.

**Covariance and Robustness of ST-DNN:** To demonstrate the covariance of ST-DNN to translation and rotation, each image in the test set was randomly rotated and cropped to a rectangle at random positions (to simulate translation) in the rotated image. We segment the original and the transformed image, denoted $S[I]$ and $S[I \circ w]$, respectively, where $w$ is the transformation used to produce the translated/rotated image. We then measure the difference between $S[I] \circ w$ and $S[I \circ w]$ through GT-covering; both should be equal if the descriptor is covariant. Results are summarized in Table 1. ST-DNN outperforms resnet101-all-ce by a margin of almost 25%. Note ST-DNN uses no data augmentation, whereas the competing networks are augmented with translated and rotated images. To demonstrate robustness to deformation, for each image in the dataset we generate random deformations of varying deformation norm $\|v\|^2$ between 20 and 80. We examine the robustness of descriptors to deformations of increasing norm by comparing the segmentation of the original and deformed images similar to the covariance experiment. Results are in Table 1, which show that ST-DNN is more robust by large margins than competing descriptors, and the robustness over competing methods increases with increasing norm.

**Comparison of ST-DNN to Standard DNNs**: We compare our method ST-DNN (ours) on a texture segmentation dataset (13) against popular deep learning architectures in computer vision - DeepLab-v3 (6), and FCN-ResNet101 (11) and with a modified version of ST-DNN where the descriptor is calculated on the entire image (ST-DNN (RL)). ST-DNN networks have **8900** parameters and is trained with only **128** images. ST-DNN is around 3 orders of magnitude smaller than standard deep networks and takes around 2 orders of magnitude less training data (e.g., FCN-ResNet101 uses 50,000 images plus augmented data and has 45 million parameters), but still outperforms these networks. We have trained FCN-ResNet101 and DeepLab-v3 on saliency detection datasets (large datasets for binary segmentation) and fine tuned on the base dataset augmented with 8 rotations and 5 scales. We have used both cross entropy loss (denoted with ce) and our loss (denoted with 'ours') for standard deep learning networks in our experiments. Quantitative results are shown in Table 2. ST-DNN outperforms all other deep networks by good margins. Notice that because of the desirable properties of PDEs STDNN (RL) perform better than DNN based architecture ST-DNN (ours) performs better than ST-DNN (RL) because of the iterative update of the region.

| Deformation Experiment | | | |
|---|---|---|---|
| Sobolev Norm | 20 | 40 | 80 |
| DeepLab-v3 | 0.85 | 0.76 | 0.66 |
| FCN-ResNet101 | 0.81 | 0.75 | 0.65 |
| ST-DNN | **0.88** | **0.85** | **0.81** |

| Covariance Experiment | |
|---|---|
| FCN-ResNet101 | ST-DNN |
| 0.69 | **0.87** |

Table 1: Covariance and Deformation: Comparison of ST-DNN with DNNs

| Method | GT-cov. | Method | GT-cov |
|---|---|---|---|
| ST-DNN (ours) | **0.94** | ST-DNN (RL) | 0.89 |
| resnet101-ce | 0.79 | resnet101-ours | 0.83 |
| deeplabv3-ce | 0.86 | deeplabv3-ours | 0.85 |

Table 2: Segmentation Results: Comparison of ST-DNN with DNNs.

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

# 5 Appendix

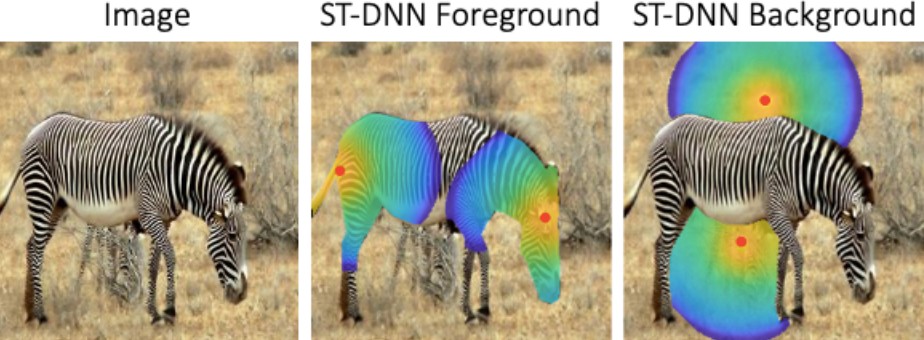

Figure 1: Shape-tailored DNNs and its optimization aim to compute dense descriptors that aggregate data only within regions of segmentation. Two different points in each of the images are shown; heat maps defined by ST-DNNs show the location's contribution to the ST-DNN descriptor at the point (yellow is high; blue is low). Points in the foreground/background aggregate only data from their respective regions (*shape-tailored*). Using Poisson PDEs to define ST-DNNs ensures it is provably robust to deformations and covariant to translations and rotations, properties that make learning more data efficient. Red points in each image both have similar descriptors as they are shape-tailored and robust/covariant, which is desired since both points belong to the same segmentation region.

## 5.1 Analytical Proofs for Covariance and Robustness

The proof of Theorem 1 (covariance of ST-DNN to rotation and translation) follows from basic properties of the Laplace equation (9); we state these properties in terms of our language of covariant operators, and show the proof for the convenience of the reader. We will show Theorem 2 (robustness of ST-DNN to domain deformations) as a consequence of properties of linear PDE theory.

We repeat the definition of covariant operator:

**Definition 2** *An operator $T : \mathcal{I} \to \mathcal{I}$ is* **covariant** *to a class $\mathcal{W}$ of transformations if*

$$T[I \circ w] = [TI] \circ w,$$

*for every $I \in \mathcal{I}$ and $w \in \mathcal{W}$.*

We show covariance of the Laplace operator, and as a consequence, the covariance of the Poisson PDE and the ST-DNN.

**Theorem 3** *The Laplacian operator $\Delta = \sum_i \frac{\partial^2}{\partial x_i^2}$ is covariant to rotations, $x \to Rx$ where $R$ is a rotation matrix.*

**Proof 1** *Let $u \in \mathcal{I}$, and let $R$ be a rotation. Consider*

$$\frac{\partial}{\partial x_i} u(Rx) = \left[ \frac{\partial}{\partial x_i} u(Rx) \right]^T \nabla u(Rx) \tag{6}$$

$$= e_i^T R^T \nabla u(Rx) = [\nabla u(Rx)]^T R e_i. \tag{7}$$

*Also,*

$$\frac{\partial}{\partial x_i} \left[ \frac{\partial u}{\partial x_j}(Rx) \right] = \left[ \nabla \frac{\partial u}{\partial x_j} \right]^T (Rx) R e_i \tag{8}$$

*so,*

$$\frac{\partial}{\partial x_i} [\nabla u(Rx)] = Hu(Rx) R e_i \tag{9}$$

*so ,*

$$\frac{\partial}{\partial x_i}\left[\nabla u(Rx)\right]^T = (Re_i)^T Hu(Rx). \tag{10}$$

*Thus,*

$$\frac{\partial^2}{\partial x_i^2}u(Rx) = \frac{\partial}{\partial x_i}[\nabla u(Rx)]^T Re_i = (Re_i)^T Hu(Rx)Re_i. \tag{11}$$

*Then,*

$$\begin{aligned}
\Delta(u \circ R)(x) &= \sum_i \frac{\partial^2}{\partial x_i^2}u(Rx) \\
&= \sum_i (Re_i)^T Hu(Rx)Re_i = tr[R^T Hu(Rx)R] \\
&= tr[Hu(Rx)] \\
&= \Delta u(Rx),
\end{aligned} \tag{12}$$

*where the second last equality is due to the invariance of the trace to similarity transformations.*

**Theorem 4** *The Laplacian operator is covariant to translations, $x \to x + t$, where $t$ is a vector.*

**Proof 2** *We have,*

$$\frac{\partial}{\partial x_i}[u(x + t)] = \frac{\partial u}{\partial x_i}(x + t). \tag{13}$$

*Similarly,*

$$\begin{aligned}
\frac{\partial^2}{\partial x_i^2}[u(x + t)] &= \frac{\partial}{\partial x_i}\left[\frac{\partial u}{\partial x_i}(x + t)\right] \\
&= \frac{\partial^2 u}{\partial x_i^2}(x + t)
\end{aligned} \tag{14}$$

*Thus, $\Delta(u \circ T)(x) = (\Delta u) \circ T$ where $T(x) = x + t$.*

**Corollary 1** *The solution $u$ of the Poisson equation, i.e.,*

$$u(x) - \alpha \Delta u(x) = I(x), \tag{15}$$

*$u = T[I]$ is covariant to translations and rotations.*

**Proof 3** *This follows from the covariance of the Laplacian, and the identity map.*

We can now show covariance of the ST-DNN to translations and rotations.

**Theorem 5** *The ST-DNN equation (3 main paper) is covariant to translations and rotations.*

**Proof 4** *Since ST-DNNs are composition of solutions of Poisson Equations with fully connected layer across channels and non-linearity in multiple layers. A linear combination of channels and point-wise non-linear operation preserves the covariance of rotation and translations, hence the ST-DNNs are covariant to translations and rotations.*

We now show robustness of the ST-DNN to domain deformations with respect to the Sobolev norm; that is, we show that the output of a ST-DNN layer does not change much if deformed by a transformation with small Sobolev norm (a smooth transformation that has small displacement). The Sobolev measures the smoothness of the deformation, i.e., the $\mathbb{L}^2$ norm of the deformation and the gradient of the deformation. We have the following theorem:

**Theorem 6** *The solution of the ST-DNN is robust to deformations, i.e.,*

$$|T[I \circ w] - T[I]| \le C\|w - id\|_{H^1}, \tag{16}$$

*where $T$ is the mapping from the image to the solution of the ST-DNN, $w : \Omega \to \Omega$ is a domain deformation, id is the identity map, and $H^1$ indicates the Sobolev norm.*

**Proof 5** *This is a consequence of Lemma equation 1 below, which shows that each layer of the ST-DNN is robust. Stacking such layers preserves the robustness by applying Lemma 1 successively.*

We now prove robustness of layers of ST-DNN. For convenience in the proof, we assume the input operates on the domain $\Omega = \mathbb{R}^2$, which avoids having to consider the boundary and more complicated formulas that do not affect the essence of the argument. Let

$$f[I] = r[(L \circ D \circ K) * I] \tag{17}$$

be a layer of ST-DNN. Here $r$ is the rectified linear unit, $K$ is the kernel representing, the Green's function of the Poisson equation (we assume for this proof that the domain of the image is all of $\mathbb{R}^2$), $D$ is the derivative operator representing oriented gradients (or parial derivatives of an finite order), and $L$ is a weight matrix of fully connected layer across channels.

We state the robustness of a layer of ST-DNN:

**Lemma 1** *A layer, $f[I] = r[(L \circ D \circ K * I]$, of a ST-DNN is Lipschitz continuous with respect to diffeomorphisms in the Sobolev norm, i.e.,*

$$|f[I \circ w] - f[I]| \leq C\|w - id\|_{H^1}, \tag{18}$$

*where $id(x) = x$ is the identity map, and $C$ is a constant (independent of $w$ and only of function of the class of images), and $\|w\|_{H^1}^2 = \int_\Omega (|w(x)| + |\nabla w(x)|^2) dx$. Note that $w - id$ is the displacement.*

**Proof 6** *Let $w$ be a smooth diffeomorphism. Then by Lipschitz continuity of the ReLu,*

$$|f[I \circ w](x) - f[I](x)| \leq |M * (I \circ w)(x) - M * I(x)| \tag{19}$$

*where $M = L \circ D \circ K$. Note that by a change of variables,*

$$M * I(x) = \sum_y M(x - w(y)) I(w(y)) \det \nabla w(y). \tag{20}$$

*Note that the determinent of the Jacobian appears if we weight the sum by the area measure, which approximates the integral. Therefore,*

$$M * (I \circ w)(x) - M * I(x) =$$
$$\sum_y [M(x - y) - M(x - w(y)) \det \nabla w(y)] I(w(y)). \tag{21}$$

*We let $w(y) = y + v(y)$. This gives us*

$$\det \nabla w(y) = 1 + div(v(y)) + \det \nabla v(y).$$

*We may bound the second term as*

$$|div(v(y)) + \det \nabla v(y)| \leq C_1 |\nabla v(y)|^2$$

*by basic inequalities. Therefore,*

$$|M(x - y) - M(x - w(y)) \det \nabla w(y)| \leq$$
$$|M(x - y) - M(x - w(y))| + C_1 M(x - w(y)) |\nabla v(y)|^2. \tag{22}$$

*By Lipschitz continuity of the Poisson kernel and derivatives, we have*

$$|M(x - y) - M(x - w(y))| \leq C_G \|L\|_\infty |v(y)|. \tag{23}$$

*Note that the Poisson kernel has a singularity at the origin, so the statement is not precise; however, as common in PDE analysis, as we will below compute integrals of the left hand quantity, we can break the integral into two terms one that integrates the singularity in a small ball (which is finite) and the other that integrates the right hand side, that we analyze below. The former will disappear to zero as the ball goes to zero. We omit the details to avoid hiding the main argument.*

*We also have that*

$$|M(x - w(y))| \leq C_2 \|L\|_\infty. \tag{24}$$

*Therefore,*

$$|f[I \circ w](x) - f[I](x)| \leq$$
$$C\|L\|_\infty \|I\|_\infty \int_\Omega (|v(y)|^2 + |\nabla v(y)|^2) dy \tag{25}$$
$$= C\|L\|_\infty \|I\|_\infty \|w - id\|_{H^1}^2.$$

For a multi layer multiple layers network with $N$ layers we will have:

$$|T[I \circ w] - T[I]| \leq$$

$$(\prod_{i=1}^{N} C_i \|L_i\|_{\infty}) C_0 \|L_0\|_{\infty} \|I\|_{\infty} \|w - \mathrm{id}\|_{H^1}^2 \tag{26}$$

$$= C\|w - \mathrm{id}\|_{H^1},$$

---

**Algorithm 1** Texture Segmentation with ST-DNNs

---

1: Input: An initialization of $\phi_i$
2: **repeat**
3:      Set regions: $R_i = \{x \in \Omega : i = \mathrm{argmax}_j \phi_j(x)\}$
4:      Compute dilations, $D(R_i)$, of $R_i$
5:      Compute $\mathbf{F_{R_i}}$ in $D(R_i)$, compute $\mathbf{a}_i = 1/|R_i| \cdot \int_{R_i} \mathbf{F_{R_i}}(x)dx$.
6:      Compute band pixels $B_i = D(R_i) \cap D(\Omega \backslash R_i)$
7:      Compute $G_i = \|\mathbf{F_{R_i}}(x)) - \mathbf{a}_i\|_2^2$ for $x \in B_i$. $\mathbf{F}$ is evaluated from the neural network.
8:      Update pixels $x \in D(R_i) \cap D(R_j)$ as follows:

$$\phi_i^{\tau+\Delta\tau}(x) = \phi_i^{\tau}(x) - \Delta\tau(G_i(x) - G_j(x))|\nabla\phi_i^{\tau}(x)| +$$
$$\Delta\tau \cdot \beta\kappa_i|\nabla\phi_i^{\tau}(x)|. \tag{27}$$

9:      Update all other pixels as

$$\phi_i^{\tau+\Delta\tau}(x) = \phi_i^{\tau}(x) + \Delta\tau \cdot \beta\kappa_i|\nabla\phi_i^{\tau}(x)|.$$

10:     Clip between 0 and 1: $\phi_i = \max\{0, \min\{1, \phi_i\}\}$.
11: **until** regions have converged

---

