# OpenReview forum: "Shape-Tailored Deep Neural Networks With PDEs"
_NeurIPS.cc/2021/Workshop/DLDE — DLDE Workshop -- NeurIPS 2021 Poster_

### Official Review · Reviewer_xz4S · 2021-10-03
**Interesting PDE based layer for robust convolutions.**

**Confidence:** 3

**Review:**

### Summary

The article proposes to take advantage of a PDE description of a segmentation problem to drastically reduce the amount of parameters (x10^3) to learn, still improving the final performance.

### Comments

**pros**: The methodology is new, drastic reduce in amount of parameters with apparent good results for similarity detection, but not clear results on actual segmentation.

**cons**:

L165: The sentence clarity can be improved.

Table 2: The metric reported GT-covering is not defined, only presented. It would be important to see the actual performance on the dataset, since GT seems to capture the similarity of the representations. Maybe the network is making great representations for similarity but not for segmentation. The paper robustness would benefit from reporting means and std of several runs with different initialization seed.

**Score:**

4: Very good paper

---

### Official Review · Reviewer_fTpK · 2021-10-08
**Thoretically grounded but poorly tested. Very promising!**

**Confidence:** 3

**Review:**

The paper is well written and their approach is grounded in solid mathematical theory: It is fairly easy to follow.
It presents an interesting approach to deal with regions that are not trivial compositions of rectangles. Although, as they point out, this is NOT per se an innovation, they add a further layer (no pun intended) which follows from their mathematical framework:
Their approach is robust against transformations such as rotations, translations, and deformations of the domain.

They test to which extent these properties are actually inherited by the architecture and achieve state-of-the-art performances.
On the other hand, it is not clear the effort put into fine-tuning their "competitors": we all know you can hammer a model to obtain better results. BUT nothing takes away the fact that the number of trainable parameters is orders of magnitude below that of SOTA DNNs.

To conclude:
1) Interesting approach and results backed by theoretical arguments
2) Experiments could be improved, more transparency could be provided as well as a clear presentation of results and metrics used.


Issues:
1) What is GT-covering?

2) Consider moving Figure 1 in the main article. It is one of your main results: do not hide it in the Appendix!

3) Consider renaming the region R with a different symbol: There are plenty of other letters that do not remind directly to the line of real numbers or to the Rotation operator in the proofs.

4) If you give proofs, you have to decide: you give a sketch or you are fully formal and define everything. No halfway. Therefore, consider enhancing proofs. For example, what is H in Line 257? I guess it's the Hessian matrix, but maybe another reader might not do the same.

5) Line 12: Either you use Poisson Partial Differential Equation or Poisson partial differential equation, not Poisson Partial differential equation.
6) Line 34: Insert a space after the dot in "consideration.3) We"
7) Make sure you use parenthesis whose height can adapt.

**Score:**

2: Borderline paper

---

### Official Review · Reviewer_J6mc · 2021-10-12
**Interesting approach. Shows promise in multiple aspects**

**Confidence:** 1

**Review:**

Summary: Author(s) propose an interesting approach to do shape-tailored spatial filtering and encode equivariance constraints by using solutions of specifically constructed poisson PDEs. This leads to a significant decrease in number of trainable parameters and amount of training data required.

Pros:

- The paper seems to largely well-written and includes appropriate references
- Encouraging results are demonstrated although on a few experiments

Cons:

- Some parts about the smoothing process (L63-73) and its incorporation into the model are hard to follow for someone with less exposure to the references.
- Needs more clarity about the experiments and the metrics reported

The reviewer is not familiar with the some of the literature behind this work but the paper and idea seem to be a good fit for the workshop.

**Score:**

3: Good paper

---

### Decision · Program_Chairs · 2021-10-15

**Decision:**

Accept (Poster)

**Comment:**

The reviews were somewhat mixed, with reviewers raising some issues with the clarity of the submission, and concerns about the need for more/better experimental tests. Nonetheless, all reviewers agreed that this is an interesting approach with great promise to deliver a robust theory-backed impact of practical relevance.